# Small and Simple Systems That Favor the Arrow of Time

**DOI:** 10.3390/e26030190

**Published:** 2024-02-23

**Authors:** Ralph V. Chamberlin

**Affiliations:** Department of Physics, Arizona State University, Tempe, AZ 85287-1504, USA; chamberl@asu.edu; Tel.: +1-(480)-965-3922

**Keywords:** arrow of time, 2nd law of thermodynamics, maximum entropy, non-extensive entropy, thermal equilibrium, stable nanothermodynamics, Ising model, Einstein oscillators, Creutz model

## Abstract

The 2nd law of thermodynamics yields an irreversible increase in entropy until thermal equilibrium is achieved. This irreversible increase is often assumed to require large and complex systems to emerge from the reversible microscopic laws of physics. We test this assumption using simulations and theory of a 1D ring of N Ising spins coupled to an explicit heat bath of N Einstein oscillators. The simplicity of this system allows the exact entropy to be calculated for the spins and the heat bath for any N, with dynamics that is readily altered from reversible to irreversible. We find thermal-equilibrium behavior in the thermodynamic limit, and in systems as small as N=2, but both results require microscopic dynamics that is intrinsically irreversible.

## 1. Introduction

The second law of thermodynamics is the fundamental barrier that prevents us from going back in time. Indeed, this “2nd law” causes the unidirectional increase in entropy needed for the “arrow of time” [1,2,3,4,5,6] and the maximum in entropy needed for thermal equilibrium. Because other basic laws of physics are reversible, the 2nd law dominates our daily lives. However, most (but not all [7,8,9]) scientists believe that irreversible behavior emerges from reversible microscopic dynamics only when systems are large and complex. As an early example, Ludwig Boltzmann often sought to obtain the irreversible 2nd law from large systems of gas particles governed by reversible Newtonian dynamics, with mixed success. His goal was to justify the maximum in entropy needed for thermal equilibrium, now embodied by Boltzmann’s factor $p\propto e^{-E/kT}$, where Boltzmann’s constant (k) connects the energy (E) to an ideal heat bath at temperature T. (An ideal heat bath is effectively infinite, with weak but essentially instantaneous thermal coupling to the system [10].) Although experiments must ultimately decide the range of 2nd law behavior [11,12,13,14], the models described here have the advantage of dynamics that is easily altered from reversible to irreversible, system sizes that are precisely fixed, and entropy of the entire system (including heat bath) that can be calculated exactly at every step.

Computer simulations provide unique insight into how macroscopic behavior emerges from microscopic dynamics [15,16,17]. In fact, the first molecular dynamics (MD) simulations on a digital computer surprised Enrico Fermi and his team by showing that small anharmonic systems governed by Newton’s laws fail to reach thermal equilibrium [18]. Although MD simulations of larger systems usually exhibit thermal-equilibrium averages [19,20], when Fermi’s ideas are extended to study dynamics, many systems have energy fluctuations that diverge from standard statistical mechanics as T→0 [21]. This disconnect between Boltzmann’s factor and Newton’s laws arises when conservation of local energy overwhelms the weak coupling to the heat bath. Here, we study a similar disconnect between Boltzmann’s factor and thermal behavior in systems that are simple enough to allow exact calculation of all contributions to entropy. We find that in this simple system the 2nd law requires steps that are intrinsically irreversible, yielding the arrow of time in small and simple systems and a minimal model for maximum entropy.

## 2. Models

All systems studied here are based on the Ising model, consisting of interacting binary degrees of freedom (“spins”) on a periodic lattice [22]. We start with the one-dimensional (1D) Ising model having N spins in a 1D “ring” (to avoid anomalous endpoints). The potential energy is $U=-J\sum_{i=1}^{N}b_{i}\sigma_{i}\sigma_{i+1}$. Here, J is an interaction energy and $b_{i}$ governs whether this interaction (“bond”) occurs between neighboring spins at lattice sites i and i+1, with $b_{N}$ connecting spins at the ends of the ring (i=N and i=1). Bonded spins ($b_{i}=1$) have interaction energy −J if they are aligned or +J if they are anti-aligned, with $b_{i}=0$ if they are unbonded. In the standard Ising model, $b_{i}=1$ for all spins. Here, most simulations have $b_{i}=0$ intermittently, so that the number of non-interacting bonds is $N_{0}=N-\sum_{i=1}^{N}b_{i}≲N/2$, to ensure maximum entropy and thermal equilibrium in stable nanothermodynamics [23,24]. Intermittent $b_{i}=0$ also adds complexity via a third energy value between neighboring spins, which is needed for the arrow of time.

Thermal behavior of 1D Ising-like spins can be treated analytically for systems of any size using Boltzmann’s entropy [25]:(1)$$S_{U}/k=\ln\left[\frac{N!2^{N_{0}}}{\left(N-N_{0}-N_{x}\right)!N_{0}!N_{x}!}\right].$$

Here, $N_{x}$ is the number of +J bonds between anti-aligned spins, leaving $N-N_{0}-N_{x}$ as the number of −J bonds between aligned spins. Note that the term $2^{N_{0}}$ is a consequence of $N_{0}$ independent segments of spins (having non-bonded endpoints) in the ring, which is replaced by $2^{N_{0}+1}$ when there are free boundaries or when $N_{0}=0$ [26]. However, the likelihood of having no broken bonds ($N_{0}=0$) is negligible if N≫10 and T is not too low because the system rarely fluctuates far from the $N_{0}/N=0.5$ value that maximizes the entropy of mixing. At the energies investigated here, equilibrium values are: $N_{0}/N=0.43-0.49$. In terms of the bond types appearing in Equation (1), the potential energy becomes:(2)$$U/J=-(N-N_{0}-2N_{x}).$$

We study 1D Ising-like spins by adapting the Creutz model [27]. In the Creutz model, Ising spins exchange energy with an explicit heat bath of “demons” (Einstein oscillators) having the same energy spacing between discrete levels. In our adaptation, there is one demon fixed to each spin site. The demon at site j serves as a local source of energy, $K_{j}=Jn_{j}$, where the number of energy units in the demon ($n_{j}$) is a non-negative integer, yielding $K_{j}\geq 0$ that is characteristic of a kinetic energy. Summing over all sites gives $K=\sum_{j=1}^{N}K_{j}$. The rules of the model ensure that total energy (E=U+K) is always exactly conserved.

The Creutz model utilizes the dynamics of cellular automata [28] to yield Monte-Carlo (MC) thermal statistics from a microcanonical simulation [29]. Cellular automata are useful as [30] “(i)…*computational tools*” for the efficient study of simplified models, “(ii)…fully discrete *dynamical systems*… (that) are relevant to physics only so far as dynamical systems are relevant to physics”, or “(iii)…original *models* for actual physical phenomena, possibly competing with existing continuum models”. An example of (iii) is [31]. Our Creutz-like model is (i) useful as a computational tool but may also be (ii) as relevant to physics as other dynamical systems. Like most microcanonical MD simulations, the Creutz model is assumed to reach thermal equilibrium without explicit use of detailed balance or Boltzmann’s factor. Indeed, a modified Creutz model has been found to match MD simulations of a classical fluid [32]. Of special interest is when the Creutz model is made reversible [29]. Unlike most MD simulations, where round-off errors and sensitivity to initial conditions often yield divergent trajectories [15,33], the discrete states of the Creutz model allow simulations to be run for an unlimited number of steps; then, if reversed, after the same number of steps the system returns to its exact initial state. Although many types of cellular automata also show unlimited reversibility, the Creutz model has an explicit heat bath that yields thermal statistics for small systems and thermodynamic behavior for large systems, but only if the dynamics is intrinsically irreversible.

Unlike most MC simulations that utilize a fixed temperature from an ideal heat bath, simulations of the Creutz model have fixed energy so that T is found from thermal averages. Consider N sources of K. If each $K_{j}$ was coupled to an ideal heat bath, its average energy would be given by Bose–Einstein statistics $K_{\infty }=J/(e^{J/kT}-1)$. Inverting this equation yields the average temperature: $kT=J/\ln\left(1+J{/K}_{\infty }\right)$. Although finite-size effects in small systems corrode the concept of T, the exact Boltzmann’s entropy for N sources of K can be written for systems of any size [34]:(3)$$S_{K}/k=\ln\left[\frac{\left(K+N-1\right)!}{K!\left(N-1\right)!}\right].$$

In fact, we find accurate thermal averages for N≥2 by counting the ways that E=U+K can be distributed between U and K. Specifically, the entropies Equations (1) and (3) yield the multiplicities $W_{U}=e^{S_{U}/k}$ and $W_{K}=e^{S_{K}/k}$, which have joint probabilities:(4)$$p_{U,K}={(W}_{U}*W_{K})/\sum_{U,K}{(W}_{U}*W_{K}).$$

Table 1 gives key properties of the N=2 system having total energy E/NJ≡1. Even in a relatively short simulation, all possible spin states and bonds are likely to occur, with weighted averages given in the bottom row of Table 1. Similar tables can be constructed for systems having other sizes and energies, but such tables become unwieldy for large systems.

## 3. Results

We simulate Creutz-like models as described in Appendix A. The main part of Figure 1 shows the total entropy density as a function of time for various simulations of a system having $N=2\times 10^{6}$ and E/NJ≡1. Figure 1A–C shows, respectively, the behavior at the beginning, near the middle (vertical scale $\times 10^{5}$, with a common offset), and the end of the simulations. The upper three sets of data in Figure 1A,C also have an expanded vertical scale (${\times 10}^{2}$, with a common offset). Two sets of simulations are reversible. The first set (green and grey lines) has dynamics of each step that is governed by the nearest source of K (“local K”). The second set (blue and cyan lines) has sources of K located throughout the lattice, which are chosen using randomly-ordered (but fixed) arrays (“global K”). The other sets of simulations (red and magenta lines) are irreversible, with global K chosen using new random numbers for every step. Main colors (green, blue, and red) come from the initial simulation, with spins always starting in the same low entropy state. Secondary colors (grey, cyan, and magenta) come from averaging three subsequent simulations of each type, alternating between reversible and irreversible dynamics, with spins starting in the final state of the previous simulation. All reversible simulations (green, grey, blue, and cyan) start and end in the same state, whereas irreversible simulations (red and magenta) show no tendency to return to their initial state. Horizontal lines in Figure 1B show that when averaged over all intermediate steps, entropy increases slightly with increasing randomness for the initial simulations (solid), while subsequent simulations (dashed) show a sharp increase from dynamics that is reversible (grey and cyan), to irreversible dynamics (magenta). Black lines in Figure 1A,C come from the initial simulation of a similar system, with global K and reversible dynamics but no broken bonds. The peak entropy densities of all other simulations are about kln⁡(2) higher, as is expected from the entropy of mixing when broken bonds are allowed. Such large increases in entropy indicate that the 2nd law favors intermittent interactions between particles, consistent with the thermodynamic heterogeneity measured in most types of materials [35,36,37].

Four lines in Figure 1A (black, green, blue, and red) show that the entropy during each initial simulation rises sharply from an initial state, with rates that increase with increasing randomness. Specifically, for the entropy to reach 95% of the maximum value, irreversible dynamics (red) requires a single sweep, whereas reversible dynamics requires seven sweeps with global K (blue) and eleven sweeps with local K (green); in contrast, systems with no broken bonds (black) never approach this maximum. Two lines in Figure 1A,C (red and magenta) show that systems with intrinsic randomness are irreversible. All other lines are reversible, precisely following every step back in time, even after more than $2.62\times 10^{11}$ steps.

Figure 1B displays these same simulations over an intermediate interval of times with the vertical scale $\times 10^{5}$. Although initial simulations (green, blue, and red) have considerable overlap in their fluctuations, their time-averaged entropies (solid horizontal lines) increase significantly with increasing randomness (standard errors are less than the line thickness). Furthermore, subsequent simulations exhibit a large jump in entropy from dynamics that is reversible (grey and cyan) to irreversible dynamics (magenta), which is even clearer when time-averaged (dashed horizontal lines). Thus, Figure 1 establishes that the total entropy from irreversible dynamics is significantly and persistently higher than that from reversible dynamics. Moreover, because each reversible simulation in the averaged behavior is subsequent to an irreversible simulation, spins that start in a high-entropy state evolve to lower entropy if the dynamics becomes reversible.

The inset in Figure 1 shows power-spectral densities (*PSD*) as a function of relative frequency (f) using the same sets of simulations and line colors given in the main part of the figure. Each *PSD* is found by taking the squared absolute value of the Fourier transform of $\sqrt{K}$. The two sets of irreversible simulations (magenta) often overlap, giving a measure of the uncertainty. These *PSD* decrease monotonically with increasing *f* for $10\log\left(f\right)>30$, consistent with the overdamped relaxation shown by the irreversible simulations in Figure 1A. Lines from reversible simulations having local K (grey) and global K (cyan) overlap only at high frequencies. Both have a minimum at $10\log\left(f\right)\approx 38$, then increase monotonically with increasing *f*, reaching a peak value at the highest *f*, consistent with the fast oscillations shown by the reversible simulations in Figure 1A,C. The *PSD* from reversible dynamics utilizing local K (grey) shows 1/*f*-like behavior for $10\log\left(f\right)<25$, indicative of slow energy diffusion at long times.

Figure 2A shows moving averages of the total entropy from the simulations of Figure 1, with Figure 2B,C from the separate entropies of K and U, respectively. Each symbol comes from averaging $10^{4}$ sweeps, which is positioned at the median time. Global K is used for both reversible dynamics (black squares) and irreversible dynamics (red circles). Open symbols come from initial simulations, with closed symbols coming from averaging three sets of subsequent simulations. Error bars (visible when larger than the symbol size) give the standard error from averaging the subsequent simulations. Total entropies (Figure 2A) of the initial reversible simulation (open squares) are nearly as high as the initial irreversible simulation (open circles), consistent with the same simulations in Figure 1B (blue and red lines, respectively). Entropies of subsequent reversible simulations in Figure 2A (filled squares) are sharply lower than those of irreversible simulations (filled circles), which is again consistent with the behavior in Figure 1B (cyan and magenta lines, respectively). The total entropy in Figure 2A tracks the entropy of K in Figure 2B but mirrors the entropy of U in Figure 2C, so that the reduction in entropy for reversible dynamics comes from the explicit heat bath. Thus, this behavior cannot be seen in simulations utilizing the Metropolis algorithm where the entropy of the bath is not known. Our results are consistent with MD simulations showing deviations from standard statistical mechanics [21], which is attributable to the explicit conservation of energy and intrinsic reversibility of Newton’s laws.

Now let us focus on the behavior during the middle third of Figure 2. During these middle times, all simulations have the average rate of bond-change attempts reduced to 1/10th the rate for spin-change attempts. Note that the total entropy is significantly altered only for reversible dynamics. Interestingly, this total entropy is reduced during the initial simulation but is increased during subsequent simulations. Thus, reversible dynamics yields non-equilibrium steady states with entropy that depends on the relative time scales of the dynamics and on the initial conditions [38,39]. In contrast, entropy is significantly and consistently higher for irreversible dynamics.

Symbols in Figure 2D show the time dependences of the ratio of probabilities from adjacent levels in K. Specifically, these are as follows: $\ln\left(p_{n}/p_{n+1}\right)$ with n=0 (squares), n=1 (circles), n=2 (up triangles), and n=3 (down triangles). Thus, if Boltzmann’s factor can be used for K, each symbol gives ∆K/kT, where $∆K=K_{n+1}-K_{n}=J$. Overlapping red symbols indicate that a single T applies to all levels only for irreversible dynamics of large systems. Black symbols show that reversible dynamics requires one value of effective T for n=0 and n=2, and another value for n=1 and n=3, with additional values for middle times when bond-change rates are reduced. The concept of a single T also fails for irreversible dynamics in smaller systems (N=128, green symbols), but such finite-size effects are expected when a fixed total energy is insufficient to thermally occupy higher energy levels. Thus, Boltzmann’s factor applies only in the thermodynamic limit, and only if the dynamics is intrinsically irreversible.

Figure 3 displays various entropy densities, and their differences, as a function of N. The main plot has logarithmic axes with $2\leq N\leq 2\times 10^{6}$, while the inset has linear axes with 2≤N≤16. Line and symbol colors identify the total energy during each simulation (see legends). The uppermost lines (dash-dotted) show that $S_{irr}/Nk$ increases with increasing N, approaching a constant value at large N. Symbols in the inset show that $S_{irr}/Nk$ (circles) from simulations coincide with $S_{theo}/Nk$ (squares) from summing the entropies of all states weighted by their multiplicities, similar to those given in Table 1. The inset also shows $\left(S_{irr}-S_{theo}\right)/S_{theo}$ (triangles) as percentages. Although Boltzmann’s factor does not describe such small systems, if the dynamics is intrinsically irreversible, thermal equilibrium from the entropy-weighted sum over all states remains valid down to N=2. Of course, thermal equilibrium fails for N=1 because there is no randomness in choosing the single source of K.

Symbols in the main part of Figure 3 show the size dependence of the difference between entropies, ${(S}_{irr}-S_{rev})/Nk=∆S/Nk$. Symbol color and type identify the total energy and heat bath, given in the legend. Lines, from fits to the data using ∆S/Nk=s/N+c with *s* and *c* constants, show general agreement with the behavior from local (dashed) and global (solid) K. Irreversible dynamics increases the entropy, ∆S>0, except for three blue triangles (two filled and one open) missing from Figure 3 where $N=\left(1\;or\;4\right){\times 10}^{4}$ with E/NJ≡2. Having ∆S<0 can be understood from Figure 2D by the number of values of effective T needed to describe the dynamics, which changes from many values for small systems (green symbols), to two values for large reversible systems (black symbols), and to only one value for irreversible dynamics (red symbols). More importantly, even as N→2 where ∆S/Nk~0.02, the inset shows that $S_{irr}$ closely matches the calculated multiplicity, while $S_{rev}$ is several standard deviations higher. Most importantly, linear fits to all data sets at $N\geq 4\times 10^{5}$ (green dotted lines) have positive slopes (∆S/Nk increases with increasing N), which is opposite to the behavior needed for $S_{rev}$ to approach $S_{irr}$ in the thermodynamic limit.

## 4. Discussion

Three issues that could cause reversible simulations to deviate from thermal equilibrium are: non-ergodic dynamics, persistent oscillations, and lack of detailed balance. All three issues appear in the dynamics given in Table A1 (Appendix B) for the simple system of Table 1 (N=2 and E/NJ≡1). The first column of Table A1 shows five initial configurations, all with K=4 and $\sigma_{i}=1$; five other configurations having $\sigma_{i}=-1$ are equivalent and hence not shown. The numbers in square brackets below each configuration give the values of [$K_{1}|K_{2}$] governing spins $\sigma_{1}$ and $\sigma_{2}$, and their respective bonds, $b_{1}$ and $b_{2}$. Each row in Table A1 shows a full cycle of the time evolution for reversible dynamics with local K, which differs only slightly from global K since N=2. Each time sweep contains a fixed sequence of two pairs of time steps: energy $K_{1}$ applied to step 1A (attempt to change $\sigma_{1}$) and step 1B (attempt to change $b_{1}$), with $K_{2}$ applied to steps 2A and 2B (attempts to change $\sigma_{2}$ and $b_{2}$).

Figure 4 shows the time dependences of total K for all distinct configurations of the N=2 and E/NJ≡1 system with reversible dynamics, from the top three rows of Table A1. First, note that the simulations are non-ergodic, e.g., the state K=0 is reached only if the initial configuration is [4|0] (black squares). Next, note that all configurations are recurrent, returning to their initial configuration with $\sigma_{i}=1$ after periods of 20, 24, and 16 steps for $\left[K_{1}|K_{2}\right]=[4|0]$, [3|1], and [2|2], respectively. These oscillations are undamped, which is attributable to the simplicity of the N=2 system, similar to the persistent oscillations that prevent MD simulations of simple systems from reaching thermal equilibrium [18,19,20]. Larger systems with reversible dynamics have damped oscillations, as shown in Figure 1. By analogy to MD simulations of large systems showing similar oscillations [21], deviations from standard statistical mechanics arise when conservation of local energy overwhelms the necessarily weak coupling to the heat bath. We identify these oscillations, which occur only if the dynamics is reversible, as the primary source of deviations from maximum entropy in the thermodynamic limit, $N\gg 10^{4}$. Finally, note that the N=2 system does not obey detailed balance. Specifically, from counting the fraction of times each value of K appears in all cycles of Table A1, we find probabilities of 0.237, 0.288, 0.263, 0.172, and 0.040 for K= 4, 3, 2, 1, and 0, respectively; in contrast, Table 1 gives the theoretical probabilities needed for maximum entropy: $p_{U,K}≅$ 0.208, 0.333, 0.25, 0.167, and 0.042. These two sets of probabilities differ by 3–15%, which is somewhat more than the 2–4% differences in entropy shown in Figure 3 for N=2 and E/NJ≡1. All three of these issues likely contribute to the failure of reversible dynamics to reach maximum entropy. The issue of a lack of detailed balance dominates for small N. The issue of underdamped oscillations persists into the thermodynamic limit. The issue of non-ergodic behavior is most relevant to the dependence on initial conditions shown in Figure 2A. Intrinsic randomness in the sequence of chosen spins and sources of K allows flexibility in the dynamics that ensures ergodicity, detailed balance, and overdamped relaxations needed for the system to yield maximum entropy, a well-defined temperature, and thermal-equilibrium probabilities of each configuration.

Two additional differences between reversible and irreversible dynamics are the size and randomness of the heat bath. Reversible dynamics with local K utilizes a single source ($K_{i}$) for spin $\sigma_{i}$ and bond $b_{i}$, whereas global K utilizes two sources, which are $K_{j\sigma }$ for $\sigma_{i}$ and $K_{jb}$ for $b_{i}$. In both cases, to facilitate reversibility, fixed arrays of sources are used throughout the simulation. Irreversible dynamics also has two versions, one utilizing $K_{j}$ for $\sigma_{i}$ and $b_{i}$ and the other having $K_{j\sigma }$ for $\sigma_{i}$ and $K_{jb}$ for $b_{i}$, with all sources chosen randomly each time. Although the probabilities that the same sources govern subsequent attempts to change $\sigma_{i}$ and $b_{i}$ differ greatly, 1/N  versus $1{/N}^{2}$ respectively, both irreversible versions give equivalent behavior. Thus, the key feature in the bath that causes reversible dynamics to deviate from thermal equilibrium is that every attempt to change $\sigma_{i}$ and $b_{i}$ is governed by the same source(s) of K. Note that similar differences in the size and randomness of the heat bath arise in MD and MC simulations. Specifically, particles in MD simulations usually interact with a finite set of nearby neighbors that tend to continue their interactions for several steps; in contrast, particles in MC simulations receive thermal energy from an ideal heat bath that is effectively infinite, with zero chance that the bath will be correlated over subsequent steps. Indeed, at least in the thermodynamic limit, our Creutz-like model shares some similarities with other types of simulations: reversible dynamics yields deviations from standard statistical mechanics similar to MD simulations [21], whereas irreversible dynamics yields Boltzmann statistics consistent with MC simulations.

Intrinsic randomness for the 2nd law in our simulations comes from a random number generator. We speculate about possible sources of this randomness in real systems. One possibility is that local sources of K have weak but effectively instantaneous coupling to an ideal heat bath, but this mechanism is inconsistent with MD simulations [21] and experimental evidence from most types of materials [37]. Another possibility is that intrinsic irreversibility and the 2nd law in real systems come from wavefunction collapse [1,3,8,9]. If an analogy can be made, in our model each step involves one spin coupling to its heat bath, so that N=2 may be related to a double-slit experiment. Then, during reversible dynamics the choice of K follows a randomly-ordered (but fixed) sequence, analogous to a priori knowledge about which slit passes the particle. In contrast, irreversible dynamics involves intrinsic randomness in the choice of K, analogous to an irreversible wavefunction collapse when a particle is measured as it passes through a slit.

## 5. Conclusions

Our simulations reveal some unanticipated results regarding the thermal behavior of simple systems. Intrinsically random steps are needed for maximum entropy in systems of all sizes. Specifically, intrinsically random steps are needed to validate Boltzmann’s factor and yield a well-defined temperature in the thermodynamic limit $N\gg 10^{4}$. Similarly, systems as small as N=2 can be accurately described by the concept of entropy for a weighted sum over all states, but only if there is intrinsic randomness in the choice of K. One mechanism contributing to deviations from standard statistical mechanics for reversible dynamics is that conservation of local energy can overwhelm any weak coupling to an idealized heat bath, as it does for fluctuations in MD simulations [21]. If the behavior of such simplistic simulations can be extended to more-realistic systems, our results suggest that intrinsic local randomness may be needed to yield maximum entropy for the 2nd law of thermodynamics and irreversible dynamics for the arrow of time.

## Figures and Tables

**Figure 1 entropy-26-00190-f001:**
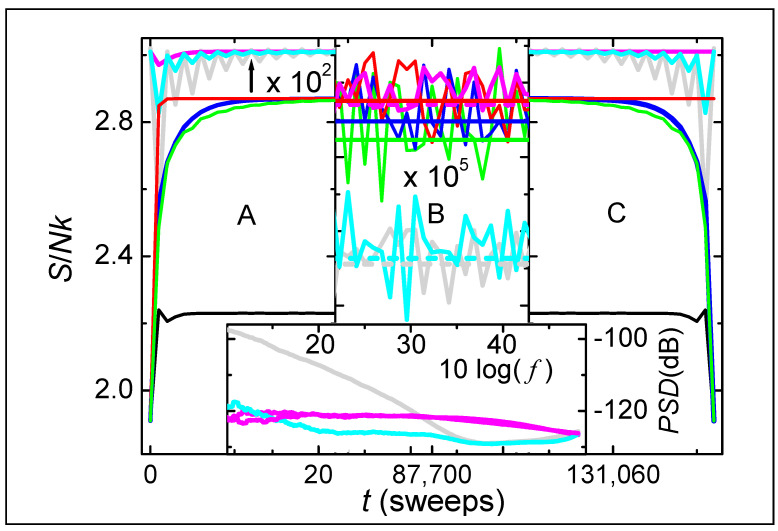
Main figures show entropy density as a function of time at the beginning (**A**), ending (**C**), and intermediate (**B**) times of simulations having $N=2\times 10^{6}$ and E/NJ≡1. Line color identifies the types of dynamics and heat bath for initial simulations (main axis) and when averaged over subsequent simulations on an expanded scale of $\times 10^{2}$ in (**A**,**C**) and $\times 10^{5}$ in (**B**), with a common offset in each case. Specifically, red (magenta) lines show initial (averaged) behavior of irreversible simulations. Initial (averaged) reversible dynamics using local K is shown by the green (grey) lines, whereas dynamics using global K is shown by the blue (cyan) lines. Horizontal lines in (**B**) show time-averaged values of each type. Black lines in (**A**,**C**) are also from reversible dynamics using global K, but with no broken bonds. The inset shows power-spectral densities as a function of relative frequency on a double-logarithmic plot, with line color matching that in the main figures.

**Figure 2 entropy-26-00190-f002:**
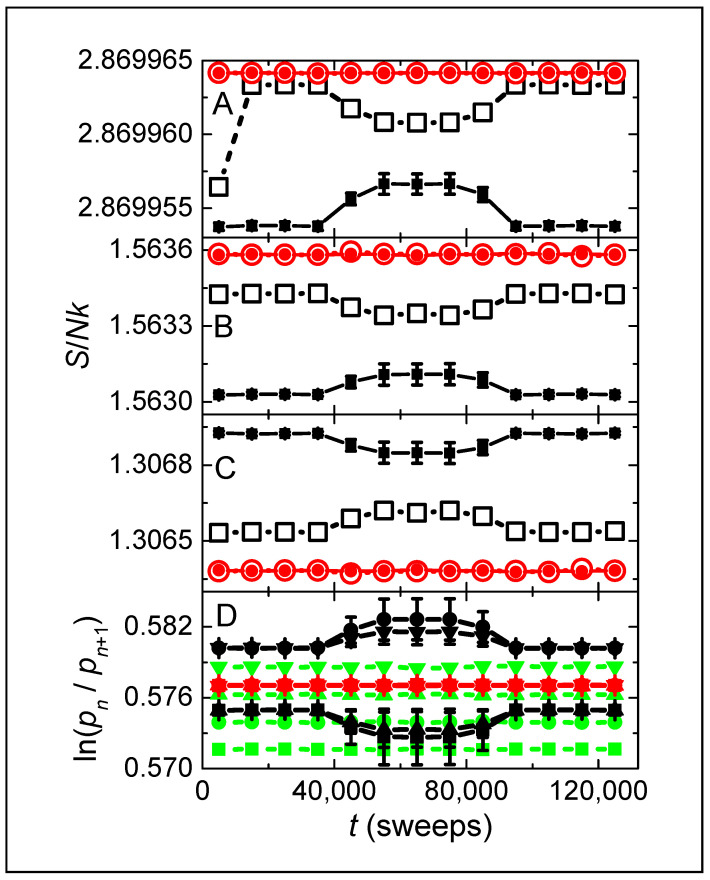
Moving averages of entropy density from the simulations shown in Figure 1 for spins (**C**), bath (**B**), and combined (**A**). Symbol style identifies dynamics as reversible (black squares) or irreversible (red circles), from initial simulations (open) or from averaging over subsequent simulations (filled), and with error bars from the standard error. During the middle third of each simulation, the average rate of bond-change attempts is reduced to 1/10th the rate of spin-change attempts. (**D**) Moving averages of the ratio of probabilities between adjacent energy levels in K. Symbol shape identifies the levels as follows: n=0 (squares), n=1 (circles), n=2 (up triangles), and n=3 (down triangles). Black and red symbols are from the simulations shown in (**A**–**C**) having reversible and irreversible dynamics, respectively. Green symbols are from irreversible dynamics in a system having N=128.

**Figure 3 entropy-26-00190-f003:**
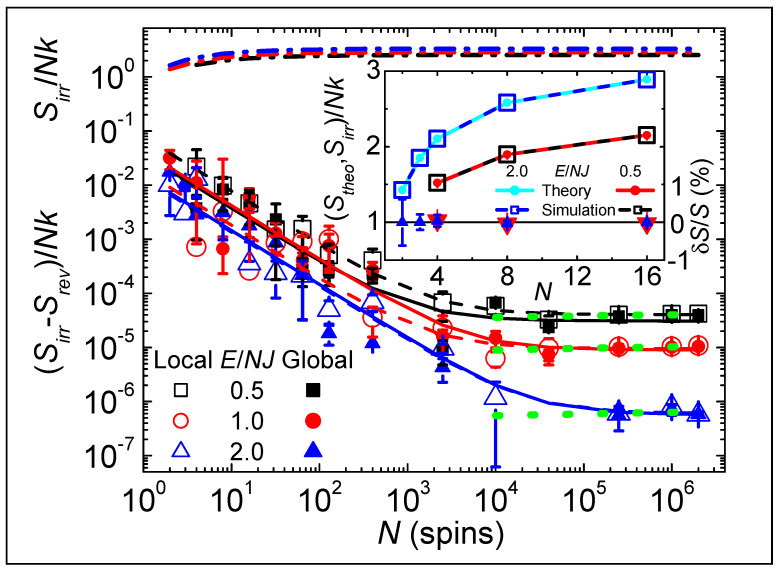
Entropies, and their differences, as a function of N on a double-logarithmic scale (main figure) and a linear scale (inset). Colors identify the total energy, as given in the legends. In the main figure, dash-dotted lines show $S_{irr}/Nk$ from irreversible simulations; symbols show $∆S/Nk=\left(S_{irr}-S_{rev}\right)/Nk$, with the bath for reversible dynamics from local K (open) or global K (filled). Error bars are from the standard error of the mean. Lines show best fits to ∆S/Nk=s/N+c for local (dashed) and global (solid) K. Dotted lines show best fits to $\log\left(∆S/Nk\right)=a+b\log\left(N\right)$ for $N\geq 4\times 10^{5}$, yielding an average of b=0.028±0.005. The inset shows behavior of small systems. Symbols give $S_{irr}/Nk$ from simulations (squares) and $S_{theo}/Nk$ from counting all multiplicities (circles). Triangles show the percent difference between theory and simulation, $(S_{irr}-S_{theo})/S_{theo}$, using the right-hand scale, with colors corresponding to energies given in the legend.

**Figure 4 entropy-26-00190-f004:**
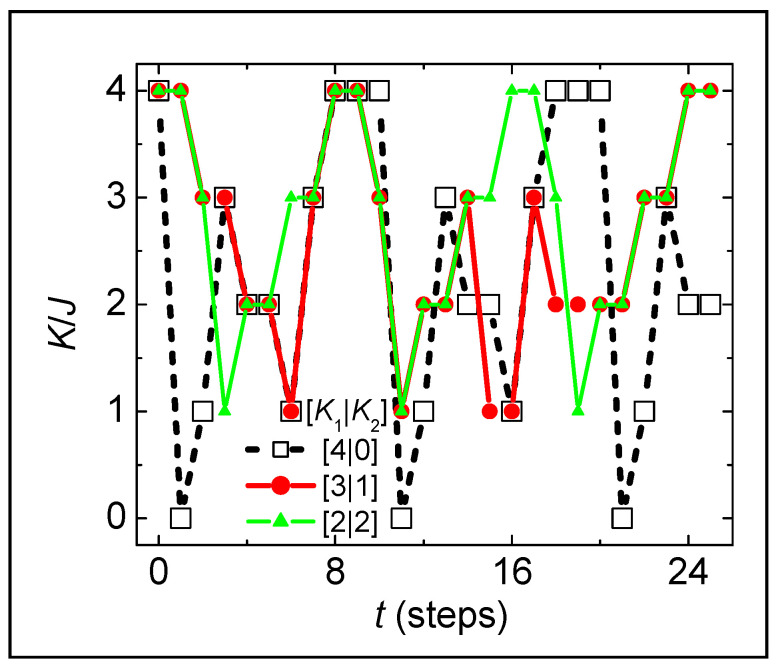
Total kinetic energy as a function of time from reversible simulations of three initial distributions of K, given in the legend, corresponding to the first three rows in Table A1 (Appendix B).

**Table 1 entropy-26-00190-t001:** Properties of an N=2 system having total energy E/NJ≡1. The first column shows representative configurations for low-energy (●), high-energy (×), or no-energy (⊙) bonds; with spins that may be up (⇑) or down (⇓). The next four columns give the number of no-energy ($N_{0}$) and high-energy ($N_{x}$) bonds, U/J (Equation (2)) and $S_{U}/k$ (Equation (1)). The next two columns give K=E−U and $S_{K}/k$ (Equation (3)). The final two columns give $S_{E}=S_{U}+S_{K}$ and $p_{U,K}$ (Equation (4)). The bottom row gives weighted averages.

Configuration	$N_{0}$	$N_{x}$	U/J	$S_{U}/k$	K/J	$S_{K}/k$	$S_{E}/k$	$p_{U,K}$
⇑●⇑●	0	0	−2	ln⁡(2)	4	ln⁡(5)	ln⁡(10)	10/48
⇑×⇓×	0	2	2	ln⁡(2)	0	ln⁡(1)	ln⁡(2)	2/48
⇑●⇑⊙	1	0	−1	ln⁡(4)	3	ln⁡(4)	ln⁡(16)	16/48
⇑×⇓⊙	1	1	1	ln⁡(4)	1	ln⁡(2)	ln⁡(8)	8/48
⇑⊙⇓⊙	2	0	0	ln⁡(4)	2	ln⁡(3)	ln⁡(12)	12/48
	Average/*N*		−0.25	0.606504	1.25	0.593788	1.2002914	

## Data Availability

The original contributions presented in the study are included in the article. Further inquiries can be directed to the author.

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
