# Peer review of "Small and Simple Systems That Favor the Arrow of Time"

_entropy, 2024, doi:10.3390/e26030190_

Round 1

Reviewer 1 Report

Comments and Suggestions for Authors

The author describes a model of spins on a 1D lattice coupled to Einstein oscillators.

The thermodynamic behaviour of the model is analysed using a set of numerical simulations, and conclusions are drawn about the evolution of entropy and the second law of thermodynamics.

It is unclear to me whether this work is a conference proceedings (putting into writing work that has previously been presented and discussed with peers at a conference) or a regular article (presenting unseen results, and requiring greater scrutiny before publication)?

In either case, I believe the presentation in the current manuscript is misleading in various aspects, and the conclusions are not sufficiently supported by the results presented in the main text.

If the author can substantially revise the manuscript so as to present clearly what is actually calculated, and to remove unsupported speculations and interpretations, I believe it may still become publishable (at least as conference proceedings). In the current form, however, I cannot recommend publication.

In particular:

1) The author performs a numerical calculation without ever specifying what type of calculation.

From the appendix, I surmised it is a type of Monte Carlo simulation, but seemingly without any detailed balance condition being implemented.

These things should be explicitly specified and the reason for not implementing detailed balance should be explained.

This is especially important because without detailed balance, Monte Carlo will typically not converge to thermal equilibrium, and different ways of constructing Monte Carlo sweeps will typically yield different ensemble averages.

It is thus unclear at the moment whether the observations by the author of the difference between what they call "reversible dynamics" and "irreversible dynamics" is due to an actual difference between different types of dynamics, or merely results from an unphysical difference in the construction of Monte Carlo sweeps.

2) There is a disconnect between the argumentation for the model being considered (any size system, explicit oscillators acting as bath, bath size proportional to system size) and the actual calculations being performed.

The definition of the system in terms of separate energies for the spins and oscillators, without any coupling, does not allow for dynamics.

Any spin + oscillator state specified in the way presented by the author, is an eigenstate of the combined spin+oscillator energy and will be static in the absence of external influences.

The "dynamics" studied in the calculation comes purely from Monte Carlo updates. If the resulting Monte Carlo averages are to converge to averaged system properties, the system must be coupled to an external bath providing a mechanism for spin and oscillator dynamics.

The actually performed calculation thus implicitly assumes the presence of an infinite bath (providing dynamics, if not an influx of energy), which is precisely the thing the author set out to avoid in their introduction.

The conclusions of the author related to the system size and the absence of an infinite heat bath are thus at best not straightforward to interpret.

3) The notion of reversibility used by the author is similarly flawed. The system as specified does not have any dynamics (all states are static), and therefore no notion of reversibility. The Monte Carlo simulation does have (Monte Carlo) dynamics, and the simulation (not the physical system) could therefore be endowed a notion of reversibility. In this case however, "reversibility" simply means storing all discrete steps the system has taken and then reversing them, whereas "irreversible" dynamics is made irreversible only by not storing the exact sequence of discrete steps. The irreversibility is thus a practical choice, and not inherent even in the numerical dynamics.

This casts doubt on the interpretation of the role ascribed to (ir)reversibility in the discussion of the numerical results.

4) The speculation in the discussion section about "wavefunction collapse" does not make sense in the context of the article. The system as defined by the author is purely classical (only commuting observables), and the Monte Carlo simulation is similarly classical (evolution in phase space, not Hilbert space). Since there is no quantum anywhere in the model or its simulation, the observed dynamics should not require any quantum effects.

Author Response

I thank the reviewer for his/her careful reading of my manuscript, and for the many helpful comments and questions. I have addressed the questions in the attached file.

Reviewer 2 Report

Comments and Suggestions for Authors

This paper reports the results of simulations of a situation in which a chain of spins is coupled to a series of oscillators. The question is whether the spin chain will display thermal behavior (as indicated by the value of its entropy). In the case of reversible interactions substantive deviations from thermal behavior are found; if an element of irreversibility is introduced (via randomness) higher entropy values are attained and thermal equilibrium is better approximated. The authors submit that these results suggest that the thermal behavior seen in practice may be due to quantum randomness related to the collapse of the wave function. 

I think that the simulation results are interesting and deserve publication. I found the description of the experiment a bit unclear in places. For example, in lines 64-70 remarks are made about the factor 2^n0. It seems to be silently assumed here that adjacent pairs of non-interacting spins cannot (or will not) occur. I found the invocation of probability arguments in this passage also puzzling, because in the description reference is only made to deterministic Newtonian laws. 

A bit later, K is introduced as the total amount of oscillator energy. But in the rest of the paper K seems to be used often as an index referring to oscillators, without explanation. I found this confusing.

It is repeatedly said in the paper that in reversible processes a system finally returns to its original state. I did not understand what was meant here. Obviously, reversible processes in general hardly ever return to their initial states (think of a Newtonian particle in inertial motion). Perhaps the authors think of Poincaré recurrences in a finite phase space volume? But that depends on determinism, not on reversibility. Or is the simulation algorithm such that it retraces its steps from a certain point in time? I could not find a clear explanation of this in the description of the simulation.

So, I think more explanation is needed to improve the transparency of the paper´s argument.

Finally, in their conclusion and the section before the conclusion the authors emphasize the possibility of a connection with quantum randomness. They do not provide detailed argumentation and I did not find the suggestion persuasive.  There are clear differences between everyday situations and the experiment simulated here, differences that can be understood completely within classical physics.  In the simulation the system and its environment are of roughly equal sizes and together form a closed system in which energy is conserved. In actual practice, however, thermal systems are in contact with an environment with an enormously large number of degrees of freedom and energy can usually leak away to infinity. This obvious difference seems sufficient to explain the thermalization of systems in everyday situations, and I see no reason to invoke the collapse of the quantum wave function---at least, additional argument on this point seems necessary.

Author Response

I thank the reviewer for his/her careful reading of my manuscript, and for the many helpful comments and questions. I have addressed the questions in an attached document.

Round 2

Reviewer 1 Report

Comments and Suggestions for Authors

I thank the author for the replies to my comments and for the changes made in the manuscript.

I'm afraid, however, that these answers and changes do not alleviate my worries about the way the simulation is done and the conclusions that can be drawn from them.

1) I thank the author for specifying that the simulation is a probabilistic cellular automaton, which has characteristics of MD as well as MC calculations.

However, this does not address the observed difference between "reversible" and "irreversible" dynamics being made purely at the level of the automaton updates. In fact, the author themselves now agrees that the differences in the way the reversible and irreversible update rules are defined affect the dynamics. These evolutions can therefore not be compared in equal footing. Moreover, if only the way of simulating is different between "reversible" and "irreversible" dynamics, and not the system being studied, then any observed differences in physical quantities must be due to differences in the way of simulating, rather than any sort of physical effect.

2) I thank the author for explaining that the dynamics in the simulation is defined to be a cellular automaton. However, this only emphasises the problem: the actual physical model system being studied has no dynamics. The simulation does, based on automaton rules that do not represent any elements of the model system as introduced by the author. The automaton update rules thus implicitly introduce physical influences beyond the spins and oscillators defined in the model. Put more strongly: There is no need to simulate the dynamics of the spin+oscillator model at all. It is solved analytically in a trivial manner, because it has no dynamics. All spin+oscillator configurations are static eigenstates of the Hamiltonian as introduced by the author. The "simulation" does not agree with the analytical solution, because it implicitly introduces dynamics that is not included in the model itself. This additional, implicit ingredient is a coupling to an unspecified, infinite (memory-less) bath providing energy-conserving fluctuations.

3) I thank the author for their explanation, but the issue is the same as under 2: the "dynamics" being studied is that of the update sweeps, which are inconsistent with the model being studied by itself. Cellular automata are useful tools in physics, but only if their dynamics is used to approximate or represent the physical model being studied. In this case, the update rules of the automaton do not come from the sysem as defined by the author, and must represent influences from an infinite bath,. Moreover, the differences in update rules between "reversible" and "irreversible" dynamics must represent two different types of baths.

4) I thank the author for rephrasing this section.

Author Response

I have addressed the questions of Reviewer 1 in the attached file.

Reviewer 2 Report

Comments and Suggestions for Authors

The revision has improved the paper. I still think that the relation between the described simulation and everyday situations is unclear and that the speculation about the role of quantum randomness is unpersuasive, but in my opinion the present version of the paper may give rise to discussion.    

Author Response

I have addressed the concerns of Reviewer 2 in the attached file.

Round 3

Reviewer 1 Report

Comments and Suggestions for Authors

I thank the author for their reply and updates to the article.

The difference between the definitions of reversible and irreversible dynamics is now more clearly described.

I still do not believe the automaton update rules actually reflect the dynamics of the system defined in the introduction, but at least the fact that a cellular automaton is being used, and that its update rules are defined in a particular way, are now made explicitly clear in the manuscript.

With this, I think it is publishable as a conference proceedings.